# The Effects of *Okara* Ratio and Particle Size on the Physical Properties and Consumer Acceptance of Tofu

**DOI:** 10.3390/foods12163004

**Published:** 2023-08-09

**Authors:** Kay Hyun Joo, William L. Kerr, George A. Cavender

**Affiliations:** 1Department of Food Science and Technology, University of Georgia, Athens, GA 30602, USA; khjoo16@uga.edu (K.H.J.); wlkerr@uga.edu (W.L.K.); 2Department of Food, Nutrition and Packaging Sciences, Clemson University, Clemson, SC 29634, USA

**Keywords:** okara, tofu, sustainability, rheological property, consumer preference, texture property

## Abstract

Okara, the solid byproduct of soymilk production, poses a sustainability concern, despite being rich in fiber and other healthful compounds. In this study, the physical properties of tofu made from soymilk fortified with differing levels of okara—either whole or fine (<180 µm)—and made with the traditional coagulant nigari were examined. The yield increased linearly with the okara concentration with values of 18.2–29.5% compared to 14.5% for the control. The initial moisture in the fortified samples was higher than the control (79.69–82.78% versus 76.78%), and both the expressible moisture and total moisture after compression were also greater in the fortified samples. With a few exceptions, the texture parameters did not differ between samples. Dynamic rheology showed that all samples had G′ > G″. The storage moduli increased at different rates during each gelling step, with G′ before and after gelling increasing with the fortification level, and was greater for the samples with fine particles than with whole particles. Consumer sensory panels using the hedonic scale showed traditional tofu had a slightly higher acceptability, but the panelists indicated they would be more willing to purchase okara-fortified tofu because of the health and sustainability benefits it might have. Thus, tofu could be produced with added okara with predictable but not profound changes in its physical properties.

## 1. Introduction

Tofu has long been a significant source of protein for people in Asia, with the first evidence of tofu production recorded in the year 965 [1]. Due to concerns with sustainability and the desire to develop high-quality foods based on plant rather than animal proteins, soybean foods are again being highlighted as a protein source [2]. Soybeans and the various products made from them are well known as some of the few plant-based protein foods that provide well-balanced amino acid profiles. Reynaud et al. [3] reported that the true ileal digestibility of tofu is 95% and has a digestible indispensable amino acid score between 63 and 101%. Further, not only do soybeans compare favorably to the protein quality of animal meat, but some studies suggest they may also help reduce the health risks of metabolic problems such as cardiovascular disease, blood pressure, and diabetes [4].

Tofu production is an involved process in which hulled soybeans are ground and boiled to extract soluble proteins (glycinin (11S) and β-conglycinin (7S)) and to destroy compounds such as trypsin inhibitor and lipoxygenase, which function as anti-nutrients [5]. The resultant soy slurry is then filtered to obtain fresh soymilk and the principal by-product, okara. Next, an appropriate coagulant, commonly nigari (a seawater-derived blend of magnesium and calcium salts, primarily containing MgCl_2_), calcium sulfate, or glucono-δ-lactone, is added to the soymilk, and after a brief rest, the tofu curd is formed. Finally, these curds are loaded into molds for pressing, with the firmness of the tofu being determined by the amount of pressure and time, which are adjusted according to the purpose for which the tofu will be used.

Okara, the chief by-product of soymilk and tofu production, presents serious disposal problems given the amounts created relative to the product output [6]. According to Li, S. et al. [7], during industrial mass production, as much as 250 kg of okara is generated for every 1000 L of soymilk produced, with the worldwide generation reaching approximately 14 million tons of okara every year. In Japan alone, 16 billion JPY (about 145 million USD) is spent yearly for the disposal of the 800,000 tons of okara generated [8]. If we extrapolate those disposal costs to worldwide production figures, approximately 2.5 billion USD is spent annually to deal with the industry’s waste stream. Even after okara is disposed in landfills, severe leachate and odor issues arise, as the residual nutrients and high moisture allow microorganisms to ferment in the waste [9].

Despite being seen as a nearly worthless product, okara actually contains several important nutrients. Gupta et al. [10] analyzed okara and found high levels of fiber (52.8–58.1% of the total dry mass), significant amounts of protein (25.4–28.4%) and oil (9.3–10.9%), and a small but nontrivial amount of non-fiber carbohydrates (3.8–5.3%). In contrast, tofu has roughly one-tenth the fiber of okara (5.4%) but a much higher protein (53.9%) and oil (30.2%) content [11,12]. The difference in fiber content is of particular note, as fiber is a necessary nutrient that is often recommended in the diet of those in developed countries. Furthermore, increased consumption has been seen to have positive effects against a variety of common maladies, from constipation to cancer [13]. Thus, if some amount of the residual okara could be incorporated into tofu without loss of quality, the resultant product would higher in fiber, and by using the waste product, the process sustainability would be improved and environmental impact would be lessened.

In the current study, okara was introduced to soymilk, which was then subjected to traditional tofu manufacturing using nigari as the coagulant, thus allowing okara to become incorporated into the gel. This was done to determine if an optimal level and particle size of okara could be incorporated while maintaining product quality. Thus, the overall objectives of the current research were (1) to explore the possibility of introducing okara, a typical waste product of soymilk manufacture, into a nigari-infused tofu structure, (2) to study the physical characteristics of tofu with added okara compared to traditional tofu, (3) to investigate the effects of the particle size and the weight ratio of okara on the tofu structure, and (4) to determine consumer likeability and intent to purchase tofu with added okara.

## 2. Materials and Methods

### 2.1. Materials

Soymilk, tofu, and okara samples were prepared from whole soybeans (Glycine max) purchased from Grain Place Foods Inc. (Marquette, NE, USA). The nigari used in the process of coagulating the tofu was bought from Sanlinx Inc. (Dandridge, TN, USA).

### 2.2. Tofu and Okara Production

All steps of the process, from hulling the soybean to making soymilk and tofu, were done according to Joo & Cavender [14] and Ullah et al. [15], with some modifications (Figure 1, adapted from Cai & Chang [16]). Prior to soymilk and tofu production, the soybeans were subjected to a slight desiccation step in order to remove the surface moisture for easier hulling. To do this, 800 g of soybeans were spread out on a perforated tray until they formed a monolayer and were then dried in an impingement oven (Impinger 1450, Lincoln Inc., Fort Wayne, IN, USA) at 100 °C for 5 min. Next, a disk grinding mill (Model 4E, Quaker City Grinding Co., Phoenixville, PA, USA) set to a ~0.4 cm gap was used to crack the soybeans into two cotyledons, allowing the hulls to be separated from the kernels. Compressed air jets were then used to winnow the hulls away from the soybean kernels. The hulled samples were then placed into a freezer at −20 °C and stored there to prevent loss of quality until needed for the experiments.

Prior to producing the soymilk, ~100 g frozen soybeans were soaked overnight (12–16 h) in deionized (DI) water at 3 °C in order to thaw and aid in the separation of any remaining hulls. After soaking, any hulls floating in the water were removed from the container, and the hydrated kernels were drained of water. These drained kernels, along with 1400 g of fresh DI water, were placed into a consumer soymilk maker (Soyajoy G4, Sanlinx Inc., Dandridge, TN, USA), where they were first ground into a slurry and then heated for 30 min at 90 °C using the automated “soaked beans” function. Next, the cooked soy slurry was separated into the okara (residue) and the soymilk (filtrate) by filtration through six layers of grade 90 cheesecloth. In each replication, the resultant soymilk was divided into two aliquots with 1250 g reserved for tofu production and 30 g for the rheological analysis. The samples were kept refrigerated at 3 °C prior to the analyses up to 3 days.

The amount of okara added to each 1250 g soymilk aliquot was based on the yield of okara from each 100 g of soybeans. This yield was determined by means of a pilot study carried out to measure the average weight of freeze-dried okara produced from a batch of soymilk. Subsequently, enough okara was added to the soymilk to give the equivalent of 100%, 50%, and 25% of that yield (30.7, 15.4, and 7.7 g, respectively). The preparation of okara with differing particle sizes (whole and fine) was carried out according to the methods of Lan et al. (2020). Okara separated from soymilk was freeze-dried (Revo R&D Freeze Dryer, Millrock Technology Inc., Kingston, NY, USA) before grinding using a high-speed blender (Professional Series 500, Vitamix, Olmsted Township, OH, USA). To produce fine okara, the resultant powder was sieved with a number 80 stainless steel sieve (USA Standard Sieve Series, Newark Wire Cloth Company, Newark, NJ, USA) on a shaker (RX-29, W.S. Tyler, Mentor, OH, USA) to give a particle size of 180 μm or less, with the size distribution being further confirmed by a laser diffraction particle size analyzer (Mastersizer, Malvern Panalytical Ltd., Malvern, Worcestershire, UK). Dried and ground okara was kept sealed in a freezer at −20 °C prior to use.

A modified version of the methods developed by Cai & Chang [16,17] were used to produce tofu. Briefly, for each replication, a liquid coagulant was made by dissolving 2.8 g of nigari in 10 mg of DI water. For the experiments incorporating okara, the requisite amount of okara powder of a given size was added to the soymilk prior to coagulation. Freshly prepared soymilk samples were cooled to 60–65 °C before adding the liquid coagulant; then, the samples were stirred vigorously for 1 min. The liquid was then allowed to curdle for 1 h before being poured into a mold (10.2 × 8.4 cm) lined with three layers of grade 90 cheesecloth. When all the curd was transferred, the cheesecloth was folded over the top of the mold before the addition of the lid. Pressure was applied by means of a 1 kg weight placed atop the lid for 2 h. After pressing, the samples were demolded, the cheesecloth removed, and the resultant tofu was stored at 3 °C for up to 3 days. Three batches of samples were prepared per treatment, and all experiments were performed in triplicate per batch.

To produce a greater quantity of samples for the sensory evaluation, the process was upscaled while using the same okara ratios as the small-scale trials. Approximately 700 g of frozen hulled soybeans were used for a batch, which were soaked overnight for 16 h in DI water at 3 °C. Soybean kernels were drained and ground with 9.8 kg DI water in a stone mill (Super Masscolloider MK CA6-3, Masuko Sangyo Co., Kawaguchi, Saitama, Japan). Then, the soy slurry was poured into a steam-jacketed kettle (Groen TDC/2-20, Dover Co., Downers Grove, IL, USA) to be heated to 80–90 °C for 30 min. Okara was separated from the soymilk by filtering through 6 layers of grade 90 cheesecloth. The filtered soymilk was weighed at 8.75 kg for the control samples. An additional 53.9 g of sieved okara was added to the soymilk for the 25% fine okara sample. When the soymilk samples cooled to 60–65 °C, an aliquot of liquid coagulant (53.9 g of nigari dissolved in 70 mL DI water) was added to the soymilk, followed by 1 min of vigorous stirring. The tofu curd was allowed to coagulate for 1 h, then poured into perforated tofu molds (21 × 26 cm), which were lined with three layers of grade 90 cheesecloth. After the mold was filled, the cheesecloth was folded over the top of the curd before applying the weight to press the tofu. The 6.1 kg weight was released after 2 h, and the demolded tofu was kept refrigerated up to 3 days before use in the sensory studies.

### 2.3. Yield Determination

The tofu yield was calculated based on the mass of soymilk (1250 g), added okara (either 30.7, 15.4, or 7.7 g), and tofu using Equation (1):(1)$$Tofu\;Yield\left(\%\right)=\frac{\left(g\;tofu\right)}{\left(g\;soymilk\right)+\left(g\;added\;okara\right)}\times 100$$

### 2.4. Color Analysis

A colorimeter (Croma Meter CR-410, Minolta Co., Ltd., Osaka, Japan) was used to assess the color of the soymilk just prior to the addition of the coagulant and on the tofu after demolding and overnight storage. The colorimeter was calibrated according to the manufacturer’s specifications at room temperature using the manufacture-provided white calibration plate (Minolta Co., Ltd., Osaka, Japan; Serial # 13333105; Reference values Y, x, and y: 83.7, 0.3199, and 0.3266 or L*a*b*: 93.32, 4.74, and 0.375). All color values were evaluated in the CIE L*a*b* color space (L* = lightness, a* = redness/greenness, and b* = yellowness/blueness). For the soymilk samples, aliquots were poured to a 12 cm thickness and placed on a white background, while the tofu samples were measured directly on the surface.

The whiteness index (WI_Hunter_) [18] and yellowness index YI_FC_ [19] were computed from the L*a*b* values by Equations (2) and (3), respectively:(2)$${WI}_{Hunter}=L^{*}-3b^{*}$$
(3)$${YI}_{FC}=142.86\times b^{*}/L^{*}$$

Referring to Sharma [20], the two points in CIELAB space can determine whether the optical difference can be perceived or not through calculating the distance between the two points (ΔE* value). The ΔE* values between all products were computed and considered “just noticeably different” when the ΔE* value was around 2.3 (Equation (4)).
(4)$$\Delta E^{*}=\sqrt{{\left(\Delta {L^{*}}_{2}-\Delta {L^{*}}_{1}\right)}^{2}+{\left(\Delta {a^{*}}_{2}-\Delta {a^{*}}_{1}\right)}^{2}+{\left(\Delta {b^{*}}_{2}-\Delta {b^{*}}_{1}\right)}^{2}}$$

### 2.5. Moisture Properties

The water-holding capacity, as determined by expressible moisture, was measured as described by Matemu et al. [21]. For each analysis, a 10 g cuboid tofu sample was placed into a 50 mL conical tube layered with 90 grade cheesecloth. Three samples were prepared per block of tofu. A sample was centrifuged for 0.5, 1, 3, 5, 10, 15, 20, 25, and 30 min at 1300× *g* in a Model 5702 centrifuge (Eppendorf, Hamburg, Germany). This allowed for both a time-based and an overall assessment of water lost from the samples when subject to a force. Freed water was decanted from the conical tube after each centrifugation period. Finally, each sample was freeze-dried to determine the dried mass. The mass of the initial sample (w_1_), the conical tube and cheesecloth (w_0_), the tube with the sample after decanting the water (w_2_), and the tube with the dried sample (w_3_) were all recorded. The values were used to calculate the total initial moisture content (M_0_: Equation (5)) and the moisture content at each centrifuged time (M_t_: Equation (6)):(5)$$M_{0}\left(\%\right)=\frac{initial\;mass\left(w_{1}\right)-dry\;mass\left(w_{3}-w_{0}\right)}{initial\;mass\left(w_{1}\right)}\times 100$$
(6)$$M_{t}\left(\%\right)=\frac{centrifuged\;mass\left(w_{2}-w_{0}\right)-dry\;mass\left(w_{3}-w_{0}\right)}{centrifuged\;mass\left(w_{2}-w_{0}\right)}\times 100$$

The moisture content before and after 30 min of centrifuge were also recorded, allowing moisture loss per gram of dry solids to be computed using Equation (7):(7)$$Moisture\;loss=\frac{centrifuged\;mass\left(w_{2}-w_{0}\right)-initial\;mass\left(w_{1}\right)}{dry\;mass\left(w_{3}-w_{0}\right)}$$

### 2.6. Texture Analysis

Three cylindrical samples (15 mm in diameter by 15 mm in height) were excised from each block of tofu and subjected to a texture profile analysis (TPA) using a TA-XT2 Texture Analyzer (Stable Micro Systems, Godalming, UK) fitted with a 45 mm cylindrical probe and 5 kg load cell. The TPA settings were 75% compression, 1.00 mm/s pre-test speed, 0.50 mm/s test speed, 1.00 mm/s post-test speed, and 20 g trigger force. Force per time data were collected, and the TPA parameters of hardness, springiness, cohesiveness, resilience, and chewiness were calculated using Exponent Connect Software (Texture Technologies Corp., Hamilton, MA, USA). Figure 2 shows a representative graph of a TPA test and how the parameters are determined.

### 2.7. Rheological Properties during Tofu Gelation

A dynamic rheometer (Discovery HR-2, TA Instruments, New Castle, DE, USA) was used to determine the rheological property changes during tofu gelling. For each sample, 2.5 mL of soymilk was treated with 0.29% nigari (*w*/*w*) and loaded immediately onto a 40 mm cross-hatched base plate, which was modified with a trench around the plate to facilitate the application of mineral oil to prevent evaporation from the open sides. A 40 mm cross-hatched probe was lowered onto the sample to a gap of 1 mm. Dynamic oscillatory tests were done at 0.5% strain and a frequency of 1.0 Hz. The storage modulus (G′) and loss modulus (G″) were recorded over time across 3 distinct temperature steps. First, the temperature was increased from 25 to 85 °C with a 5 °C increase per minute. Next, the sample was held at 85 °C for 60 min before being cooled from 85 to 25 °C with a cooling rate of 5 °C per minute.

### 2.8. Sensory Evaluation

A consumer acceptability test was performed with a 9-point hedonic scale using the control and 25% fine okara samples. The evaluation was held in the sensory facility at the Food Processing Research and Development Lab, University of Georgia (UGA), using 95 panelists (35 males and 60 females). The research participants were recruited from the general population and studies conducted under the auspices of the UGA Institutional Review Board. Also, informed consent was obtained from each subject prior to their participation in the study.

Both the control and the 25% fine okara samples were cut into 15 × 15 × 15 mm cubes and were divided into 400 g batches for cooking. Each batch was pan-fried for 10 min with 20 mL of vegetable oil and braised for 10 min in a “Jorim sauce” made from 2 g minced garlic, 1.7 g sugar, 0.5 g red pepper flakes, 10 mL rice wine, 10 mL sesame oil, 35 mL soy sauce, and 60 mL water. After cooking, the samples were kept under refrigeration at 4 °C until the day of the sensory evaluation. Prior to serving, the samples were removed from refrigerated storage and allowed to warm up to room temperature. The panelists were served three cubes of each sample in a 60 mL plastic cup, and all samples were coded with 3-digit random numbers.

The sensory panels utilized individual booths with partitions and incandescent lighting. Paper ballots were used to collect general demographic information and acceptance data on appearance, flavor, texture, and overall liking. After the evaluation of acceptability, the panelists were presented with a short paragraph describing what okara is and how it might aid in sustainability and metabolic syndrome before being asked whether they would be willing to purchase the 25% fine sample.

### 2.9. Statistical Analysis

Each analytical experiment was sampled and measured three times, and all statistical analyses were carried out by JMP (ver. Pro 15.0.0, SAS Institute Inc., Cary, NC, USA). The yield, color analysis, moisture content, and texture profile analysis were subjected to one-way analysis of variance (ANOVA) and post hoc testing by the Tukey–Kramer honestly significant difference (HSD) method where appropriate. For expressible moisture, a piecewise analysis of covariance (ANCOVA) was used to investigate the slope difference of the treatment groups, and ANOVA was implemented on individual time points to verify the mean difference of the samples. Further, ANCOVA was performed to calculate and compare the percentage of tofu yield change due to different sizes of the okara particles. Finally, for the sensory evaluations, the Student’s *t*-test was carried out for acceptability data, and a binomial test versus a “chance probability” of 0.5 was executed for the market purchase question.

## 3. Results

### 3.1. Composition, Yield, and Water-Holding Capacity

In general, the tofu yield increased proportionally with the amount of added okara, with a lower limit of 14.52% yield for the control and upper limit of 28.27% yield for the samples with 100% whole particles and 29.52% yield for the samples with 100% fine particles (Table 1). The particle size of the okara appeared to have no discernible effect on the yield. For example, samples with 25% of okara supplement had yields of 17.52% (whole okara) and of 18.16% (fine okara), and these measures were not significantly different (*p* > 0.05). Interestingly, the yield increased in a strictly linear fashion in relation to the okara content. Linear regression showed that, for the whole particles, (Yield) = 0.139 × (%Okara) + 14.39 (R^2^ = 0.998), while, for the fine particles, (Yield) = 0.150 × (%Okara) + 14.45 (R^2^ = 0.999).

As seen in Table 1, the tofu samples containing okara had a greater moisture content than the control (76.78%). The moisture content increased with the level of whole okara particles, and the samples with 25% and 50% fine particles (82.70 and 82.54%) had greater MC than those with 25% and 50% whole particles (79.69 and 80.96%). Otherwise, there were no differences in MC due to particle size, with values in the range of 82.49 to 83.89% for the samples of 100% okara of different particle sizes. Of course, the retention of more moisture may partially explain why there was a greater yield from the tofu made with added okara.

Expressible (or water-holding capacity) moisture is often determined by placing a sample under a compressive load and monitoring how the moisture level changes overall. Figure 3 displays this in a dynamic sense, showing how the moisture content changed with time when subject to a 1300 g force. In general, the steepest change in moisture occurred in the first few minutes, with a more gradual decline thereafter. The moisture content after 30 min and the total loss in moisture are shown in Table 2. The samples with fine okara particles had the highest initial moisture (82.54–83.89%), as well as the highest moisture after 30 min (64.05–64.95%). The control had a significantly lower moisture at 30 min than all other samples, except those with 25% and 50% whole okara.

However, the samples with fine okara also showed the greatest overall change in moisture after 30 min of centrifugation, with between 2.937 and 3.364 g H_2_O lost per g of dry solids. The tofu with whole particles lost between 2.366 and 2.928 g H_2_O/g of solids. The nonfortified samples had the least overall change, with only 1.829 g H_2_O lost per g of dry solids. Thus, the okara particles allowed for the incorporation of greater moisture than the control, consequently giving a greater yield and likely softer texture. However, a significant portion of this increased moisture was not tightly bound, as the fortified samples lost substantially more water when compressed, which might predict consumer experiences during chewing. Despite this, the fortified samples still retained slightly more water than the control samples, even after compression.

### 3.2. Color Analysis

The measured L*a*b* values, and the calculated whiteness index and yellowness index are presented in Table 2 for both soymilk and tofu. The soymilk was lighter in the samples with added okara, with the control being the darkest (L* = 85.06) and soymilk with 100% okara being the brightest (L* = 95.42). There were no differences in L* or a* due to particle size for the samples with the same okara levels. The soymilk samples with okara had lower a* values and higher b* values compared to the control soymilk, indicating these samples were slightly more green and more yellow in hue. In addition, the samples with 100% whole okara particles had slightly higher b* (21.55) than those with 100% fine particles (20.12), indicating the former was slightly more yellow in color. Despite these differences, the whiteness and yellowness indices of the samples were mostly indifferent, with most of the soymilk samples having no significant difference in visual whiteness and yellowness, except for the whole 50% and 100% groups, which were less white and more yellow compared to the other samples.

**Table 2 foods-12-03004-t002:** Color analyses of soy milk and tofu with added okara.

Sample	Soymilk L*	Soymilk a*	Soymilk b*	Soymilk WI	Soymilk YI	Tofu L*	Tofu a*	Tofu b*	Tofu WI	Tofu YI
Control	85.06 ^d^ (0.82)	−7.67 ^a^ (0.06)	17.52 ^e^ (0.15)	32.49 ^ab^ (0.52)	29.44 ^c^ (0.16)	85.60 ^b^ (0.31)	−5.94 ^a^ (0.04)	19.56 ^d^ (0.09)	26.93 ^a^ (0.36)	32.64 ^d^ (0.50)
Whole 25%	90.09 ^c^ (0.35)	−8.51 ^bc^ (0.03)	19.63 ^cd^ (0.06)	31.77 ^abc^ (0.21)	30.93 ^bc^ (0.06)	85.63 ^b^ (0.24)	−6.30 ^b^ (0.02)	20.63 ^d^ (0.08)	23.75 ^bc^ (0.027)	34.42 ^bc^ (0.13)
Whole 50%	93.89 ^bc^ (0.43)	−8.79 ^bc^ (0.06)	20.95 ^ab^ (0.25)	29.01 ^c^ (0.59)	32.58 ^a^ (0.32)	85.37 ^b^ (0.33)	−6.37 ^b^ (0.06)	21.20 ^ab^ (0.10)	21.76 ^c^ (0.15)	35.48 ^b^ (0.08)
Whole 100%	95.42 ^a^ (0.91)	−8.13 ^ab^ (0.42)	21.55 ^a^ (0.67)	30.78 ^bc^ (1.84)	32.25 ^ab^ (0.91)	84.97 ^b^ (0.19)	−6.03 ^a^ (0.03)	21.81 ^a^ (0.34)	19.52 ^d^ (1.08)	36.68 ^a^ (0.60)
Fine 25%	90.09 ^c^ (0.54)	−8.35 ^abc^ (0.06)	18.81 ^d^ (0.08)	33.65 ^ab^ (0.35)	29.84 ^c^ (0.09)	84.72 ^b^ (0.24)	−6.29 ^b^ (0.04)	19.48 ^bc^ (0.07)	26.29 ^a^ (0.16)	32.85 ^d^ (0.08)
Fine 50%	91.86 ^ab^ (0.24)	−9.00 ^c^ (0.03)	19.93 ^bcd^ (0.09)	34.09 ^ab^ (0.10)	30.33 ^c^ (0.07)	87.47 ^a^ (0.38)	−6.76 ^d^ (0.04)	20.81 ^bc^ (0.10)	25.03 ^ab^ (0.27)	34.00 ^c^ (0.12)
Fine 100%	95.42 ^a^ (0.85)	−8.82 ^bc^ (0.10)	20.12 ^bc^ (0.22)	35.08 ^a^ (0.29)	30.11 ^c^ (0.11)	85.20 ^b^ (0.37)	−6.53 ^c^ (0.02)	20.36 ^c^ (0.12)	24.13 ^b^ (0.12)	34.13 ^c^ (0.08)

N = 63. Means (SE) not followed by the same letter within the same column are significantly different (*p* < 0.05).

The color values for the tofu did not mirror the results of the soymilk. The lightness was significantly higher in tofu with 50% fine okara (L* = 87.47), while there was no difference in the L* values amongst the other six samples. In general, the tofu samples made with okara were marginally more green and yellow than the control samples. Further, the fine okara group was greener than the comparable whole okara group at the 50% and 100% levels, but there was no difference at 25% okara. At 25 and 100% okara, the b* values were slightly greater for tofu with whole okara particles as compared to those with fine particles. The calculated whiteness (WI) was the greatest in the control (26.93) and the lowest in the tofu with 100% whole particles (19.52). In general, the WI decreased with the okara level, and the samples with fine particles were whiter than those with whole particles at the same level.

Supporting this ANOVA difference, the ΔE value showed that some of the tofu samples could be visually distinguished by the human eye (Table 3). The ΔE value is a distance between two L*a*b* values, which is considered “just noticeably different” when the value is around 2.3 [20]. According to Table 3, the tofu with 100% whole okara and 50% fine okara were visually different from the control sample. Also, the fine 50% group was noticeably different from the fine 25% and 100% samples.

### 3.3. Texture Profile Analysis

The control tofu had a measured hardness of 855 g (Table 4), and the effect of adding okara could increase or decrease that value. For example, compared to the control, the tofu with 25% fine okara was roughly 39% softer (523 g), while that with 50% whole okara was almost 54% harder (1315 g). Taken as a group, there were no significant differences in hardness between the samples with whole okara and the control, while the samples fortified with the lower levels (25 and 50%) of fine okara were less hard than those with the whole particles. Also, the samples with fine okara particles showed profound increases in hardness as the fortification increased from 25% (523 g) to 100% (924 g). It should be noted that any differences in hardness are not easily explained, as gel firmness depends on several factors. Among these are the degree and tenacity of crosslinking, the presence of particles in interstitial spaces, and/or the capacity of particles to encourage fissures in the gel. In addition, the moisture level is relevant, as entrapped water itself may resist compression; however, water may also serve to plasticize macromolecules in the matrix.

There was no difference in resilience between the control (0.175) and samples with 25% to 100% whole particles. The tofu with fine 25% okara had the highest resiliency (0.202), and that value decreased as additional fine okara was added to the tofu. The samples with 50 or 100% fine particles were less resilient than the control. While the resilience of the tofu with 25% fine particles was the greatest, the differences amongst the samples were not large. In fact, for the related property of springiness, there was no difference in value amongst any of the samples. For both chewiness and cohesiveness, the values for the control group were no different than any other treatment group. Also, the values for tofu with added whole okara were not different from each other, while the fine okara group increased in both chewiness and cohesiveness as the content of okara escalated.

### 3.4. Rheological Properties during Tofu Gelation

Figure 4 shows the typical changes in the rheological properties when nigari treated soymilk was coagulated by heating (over the 25 to 85 °C range), allowed to stay at 85 °C for 60 min, and then cooled to 25 °C. The G′ increased and G″ decreased in all samples as the temperature increased from 25 to 85 °C. At 85 °C, both the G′ and G″ gradually increased over the hour holding period, reaching a plateau after 30 or 40 min. This showed, as was expected, that the gelling process was both temperature- and time-dependent. Finally, when the samples were cooled, there was a further pronounced increase in the G′, as well as G″. The tan δ was less than 1 in all cases, indicating a more solid-like character in the tofu samples. The tan δ decreased during the first heating phase, plateaued during the hold, then increased slightly during cooling.

Table 5 shows the values for the storage modulus (G′) and tan δ (G″/G′) at time zero (T_0_), at the beginning (T_1_) and end (T_2_) of the 85 °C hold, and after cooling the gel back to 25 °C (T_3_). Initially, the G′ ranged from 5.73 to 17.2 Pa, indicating these were very soft materials. After gelation and cooling, the values ranged from 40.6 to 108.4 Pa. This showed clear evidence that, while the firmness increased as the tofu was formed, these were not very firm gels, at least when subject to shear stress. The tan δ decreased most with the initial heating, then more gradually during the holding period. This showed that the viscous component of the matrix became less prominent during the gelation process. An interesting note is that the G′ was always higher than G″ throughout the heating and cooling phases. This suggests that, even initially, some amount of structure developed. As tan δ = 1 is often taken as a marker for gelation, in this case, there was no clear temperature or time at which gelation could be deemed to have occurred, only that the existing elastic structure became more dominant and gave a higher modulus over time.

In general, the G′ and G″ values were lower in the control than the samples with okara, either with the fine or whole particles. After the heating and cooling cycle, the control gel had G′ = 40.6 Pa (tan δ = 0.17). Those with 100% whole okara particles had G′ = 73.3 Pa (tan δ = 0.16), while those with 100% fine okara had G′ = 108.4 Pa (tan δ = 0.15). The increase in G′ also depended on the concentrations for the samples with fine okara. Thus, the G′ values were 58.0, 77.5, and 108.4 Pa for the samples with 25, 50, or 100% fine okara particles. In addition, the tan δ at the end of the temperature hold was lowest in the samples with the highest okara content.

### 3.5. Sensory Evaluation

Among the 95 panelists, all 95 people responded about the general information and tofu consumption habits, 94 people replied on the likeability of both tofu samples, and 89 people answered the market purchase question. There were 60 females and 35 males in the research participants. The age distribution was the highest in the 18–25 range (45 people), followed by the 26–35 range (38 people), the 36–45 range (8 people), the over 55 range (3 people), and the 46–55 range (1 person), respectively. The country most represented was the United States (43 people), followed by South Korea (15 people), India (11 people), China (7 people), and others (19 people). Finally, the panelists’ tofu consumption was most commonly noted as 1–3 times a month (38 people) and 1–3 times a year. Fewer participants replied that they consumed tofu 1–3 times a week (17 people) or never consumed tofu before (9 people).

Figure 5 shows the likeability of each sample, as well as intent to purchase on the market. There was no difference between flavor likeability of the control sample (7.19) and the 25% fine okara sample (7.16). However, for appearance, texture, and overall likeability, the likeability scores for the control (6.43, 6.87, and 7.06) were somewhat higher than those for samples with okara (5.56, 5.43, and 6.19). Overall, all scores were above 5, suggesting the panelists tended to like both samples more than dislike them. Finally, when information on the samples and possible health or environmental benefits were provided, 74 out of 89 respondents indicated they would more likely purchase the sample with added okara.

## 4. Discussion

### 4.1. Composition

While a proximate analysis was not performed on the samples, it is important to note that the addition of okara to the soymilk will certainly alter the composition of the resultant tofu, and those alterations may explain some of the changes that were observed. To better understand the general trends in all assays, estimated proximate compositions of the various tofu samples were performed and are presented in Table 6 (calculations based on proximate analysis data from van der Riet et al. [11]). All soymilk/okara treatments produced a tofu gel, although with differing compositions. Most notably, the addition of okara resulted in reduced protein, with values ranging from 53.9 g/100 g for the control to 41.83 g/100 g for 100% okara tofu, as well as a reduced oil content, with values from 30.20 g/100 g for the control to 20.45 g/100 g for 100% okara tofu. In contrast, the total fiber increased from 5.40 g/100 g for the control to 29.63 g/100 g for 100% okara-based tofu. In addition, there was a small increase in non-fiber carbohydrates and decrease in ash. These changes were hardly surprising, given the proximate composition of the two ingredients (okara and soymilk).

### 4.2. Yield and Water-Holding Capacity

Okara is rich in fiber [11], and fiber is well known to aid in holding water [22]. Thus, it is likely that some of the increase in the water-holding capacity has to do with the increased fiber, particularly given that Ullah et al. [23] showed that purified/extracted okara dietary fiber can help retain additional water when added to tofu. Similarly, a smaller particle size may also contribute to the binding of additional water through the exposure of hydrophilic bonding sites, as Lan et al. [24] previously showed that an increased water-holding capacity occurs when a smaller particle-sized okara is introduced into tofu samples. They concluded that an increase in both the soluble dietary fiber (SDF) and surface area could have contributed to this result.

In the current study, it is abundantly clear that the okara-fortified products entrained more water and had higher yields. As the total water content was higher in the okara products, the greater moisture loss may simply be due to the presence of more water, which may be bound less thoroughly and would therefore be easier to lose. Several factors determine the water-holding capacity of foods. For example, the molecular constituents that bind water, such as proteins and polysaccharides, may have different numbers/distribution of sites available for binding and different affinities for water [25]. Thus, one might presume that the samples with okara had more fiber and that increase in fiber helped those samples retain additional water during tofu processing. It is also important to note that, in addition to bound water, water can also be entrapped within a gel matrix. Some evidence suggests that adding fiber can influence the microstructure of hydrated gels and semi-solids, potentially creating cavities where water may be trapped [26]. Thus, one possibility is that added okara creates a net increase in the average diameter of capillary spaces in the gel, meaning that, with a given centrifugal force, it would be more difficult for water to navigate the smaller capillary spaces associated with the control formulation. Similar results have been seen in soil samples, with researchers reporting that the addition of fiber altered the capillary and gravitational water holding [26].

### 4.3. Color Analysis

While there were no differences in L* amongst the majority of tofu samples, which indicates the samples were all similarly reflective, there were some differences in the whiteness index between the samples. For example, the control formulation had higher values compared to the fortified samples, and also, when comparing the fortified samples, those made with fine okara had higher values than those fortified with whole okara. This is almost certainly explained by the differences in b* values, which measure yellowness/blueness, with positive values being yellow and negative values being blue. While no samples had a negative b* value and thus had some amount of yellow, the samples which had higher whiteness indices had lower b* values, and because the whiteness index is a measure of perceived whiteness, yellowness has a negative effect on the perception of whiteness (with blueness reinforcing the perception of whiteness) [18]. This seems borne out, as the b* value was the highest in the whole okara group and higher in the fine okara group than the control, and those groups tended to have a lower whiteness index and higher yellowness index value, respectively. The most sensible explanation for the measured differences involves the fact that soy solids naturally have a yellowish hue and the well-known fact that foods with smaller particle sizes appear more white, even if strongly colored [27].

### 4.4. Texture Profile Analysis

Resilience is a measure of how well a sample works to recover its original height after an initial compression, and as shown in Table 4, the three most resilient samples were the control and the samples fortified (either with whole or fine particles) at the lowest level (25%). Further, an increased fine okara content had a negative relationship with resiliency. Several things could explain this, but perhaps the most likely is the changes in yield and water holding. Simply put, the samples that had lower resilience also had higher overall and expressible moisture, which were actually expressed during the first compression of the TPA. As the samples rebounded between compressions, this would have left collapsed voids in the structure.

While the springiness was statistically identical across all treatments, this effect might be an artifact of the test method itself. When originally developed, the TPA procedure specified a 75% compression to mimic the act of chewing [28], yet current works use values as low as 10% and as high as 90%, with most falling in the 50–75% range [29]. At higher compression levels, products can break apart and spread, and this happened with the tofu samples in this study. This phenomenon limited recovery during the mid-test pause, preventing an actual measure of the “springiness” of the curd. Thus, in this case, it may be that resilience is a better measure for assessing the internal recovery of the structure.

Finally, both the hardness and cohesiveness increased as higher amounts of fine okara were added, as did the derived value of chewiness. This phenomenon is likely related to the numerous direct bindings of water to fiber [22], as well as the inter- and intramolecular interactions involving the fiber chains and either other fiber chains or the protein matrix. This quite likely created a firmer and more cohesive structure, along with numerous isolated voids by means of imperfect packing, which would then trap water [30].

### 4.5. Rheological Properties during Tofu Gelation

At the beginning of heating, a higher okara content was associated with a lower tan δ, and this greater degree of elastic structure persisted and built, likely through heat-initiated gelation [31]. The likely formation of stabilizing disulfide bonds continued during the isothermal (85 °C) holding phase, meaning that, when the G′ plateaued after ~1 h, most proteins should have been denatured, and virtually all possible S–S covalent bonds should have formed [30]. As the cooling phase began, both the G′ and G″ values increased sharply. This might have been due to other noncovalent bonds starting to crosslink as the temperature decreased [32]. In addition, a lower temperature increased the magnitude of the viscous element so that it offered a greater resistance to deformation.

Yan et al. [33] also found that tofu samples with added okara had higher G′ and G″ values in general. Lan et al. [24] found that the gel strength of GDL tofu gels increased by using fine okara particles in the matrix. They showed that okara-free tofu had a compact homogeneous structure when viewed by SEM. When included, okara particles became intermittently mixed in the protein network, and large particles created a more fragmented heterogeneous gel. However, tofu with fine okara particles maintained a much smoother structure.

### 4.6. Sensory Evaluation

Although the acceptance on every attribute was over 5 (neither like nor dislike), the appearance and the texture were below 6 (like slightly). According to research on foods with incorporated particles [34,35], texture acceptance can be improved by decreasing the particle size, as particles larger than ~25 µm can be detected in the mouth. Also, decreasing the surface friction and increasing lubrication can help with the later stages of mastication and perceived smoothness [36]. Again, despite slightly lower likeability scores in some categories, 83% of panelists would purchase the tofu with okara. This may be due to the tendency of consumers to select products with greater perceived health or environmental benefits, particularly if there is little difference in the quality attributes or price [37], while consumers would have to rely on government-mandated nutrition panels and potentially on-package advertising regarding the environmental benefits, as these have been shown to affect consumer purchasing habits [38,39,40].

## 5. Conclusions

It is possible to form nigari-initiated tofu gels that include okara at levels of up to 100% of that found in nonfiltered soymilk. Tofu with added okara has significantly more fiber (up to 29% compared to 5% in the control) but slightly reduced protein (as low as 41.8% compared to 53.9% for the control). In addition to nutritional changes, the incorporation of the okara, which usually represents a substantial waste stream, significantly reduces the amount of the waste during tofu processing.

The removal and treatment of okara as an ingredient allowed it to be specific particle sizes, which can have an effect on the physical properties of the resulting tofu. The particle size and weight ratio of okara did affect the tofu yield, color, and ability of the matrix to capture water and influenced the rheological properties of the developing gel, as well as the final texture of the gel. The color of tofu with okara was slightly more yellow and less white, and this phenomenon increased as higher levels of okara were added and as coarser okara was used. The inclusion of okara also allowed for an increased product yield by entraining more water into the structure while gelling. Even after exposure to prolonged centrifugal force, the water contained in the okara samples was similar to conventional tofu. In general, the textural properties such as hardness (as measured by large deformation techniques) were not significantly different when okara was included in the tofu. However, the dynamic rheological techniques indicated that the samples with okara were more elastic under shear and that the storage modulus increased with the particle concentration and decreased particle size. Finally, the sensory results showed that, although the tofu with okara had slightly lower scores for texture and appearance, consumers would be more likely to purchase those products given the information on possible health and environmental benefits.

Overall, the use of limited levels of finely ground okara will allow for the creation of tofu with quality characteristics very close to that of conventional tofu, concurrently allowing for a product with increased fiber and polyphenol content while ensuring less food waste. Finally, in order to best apply the techniques developed in this study, further studies should be carried out in order to optimize the levels and particle size for the greatest consumer acceptance.

## Figures and Tables

**Figure 1 foods-12-03004-f001:**
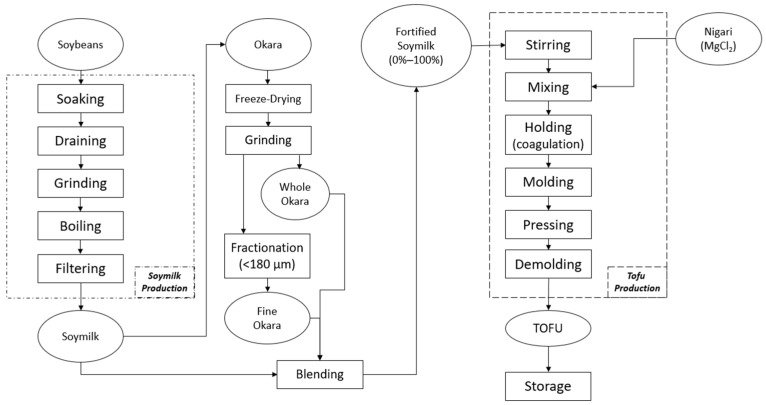
Flow diagram for the production of soymilk and tofu with fine (<180 µm) and whole okara particles.

**Figure 2 foods-12-03004-f002:**
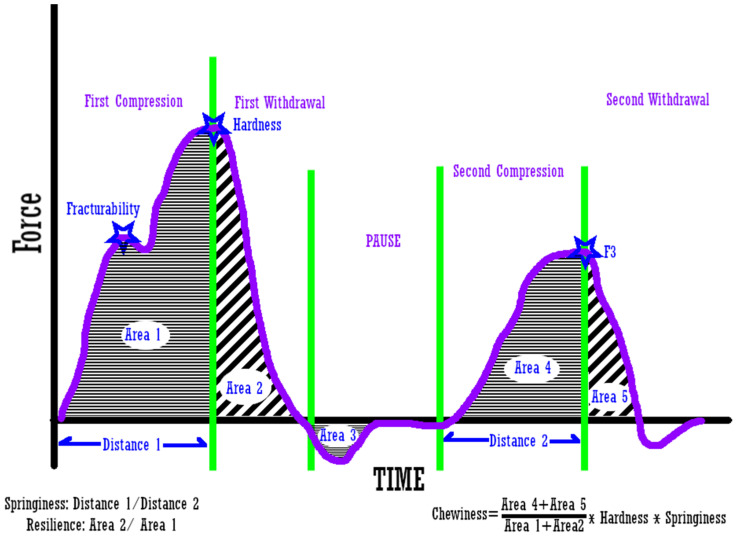
Determination of the parameters from the TPA graph.

**Figure 3 foods-12-03004-f003:**
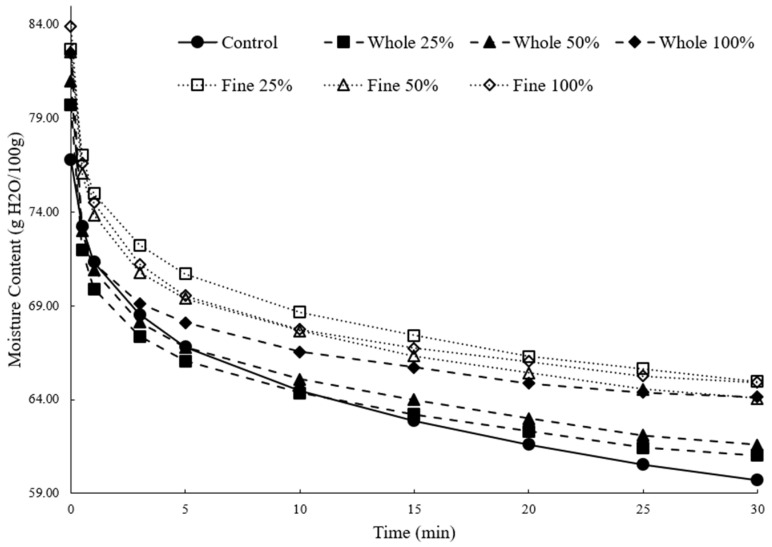
Change in moisture during compression for the control (●), 25% whole okara (■), 50% whole okara (▲), 100% whole okara (◆), 25% fine okara (☐), 50% fine okara (△), and 100% fine okara (◇).

**Figure 4 foods-12-03004-f004:**
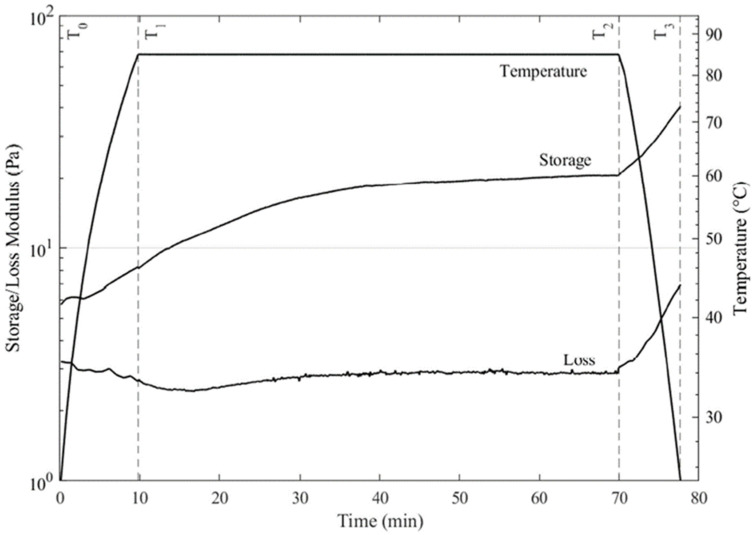
Changes in the dynamic storage modulus (G′) and loss modulus (G″) of tofu during heating from 25 to 85 °C, an isothermal hold for 1 h at 85 °C, and cooling back to 25 °C.

**Figure 5 foods-12-03004-f005:**
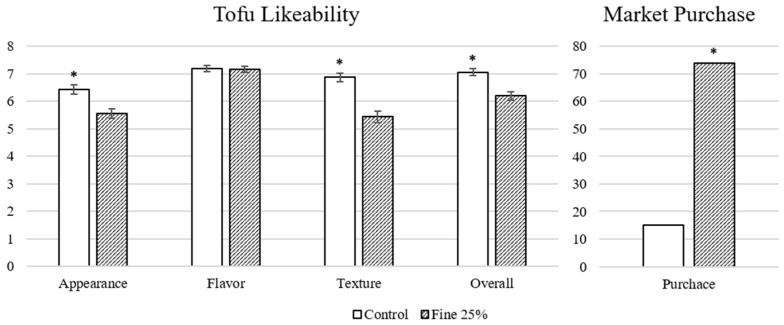
Appearance, flavor, texture, and overall likability scores and the consumers’ market purchase choice of the tofu sensory samples. Data were analyzed with the Student’s *t*-test (*n* = 94) and binomial test (X~Bin (89, 0.5)), respectively, and distinguished with an asterisk if it was significantly different (*p* < 0.05).

**Table 1 foods-12-03004-t001:** Yield and moisture content of tofu with added okara.

Sample	Yield (%)	M_0_ (g H_2_O/100 g)	M_30 min_ (g H_2_O/100 g)	Moisture Loss (g H_2_O/g Solids)
Control	14.52 ^d^ (0.11)	76.78 ^c^ (0.31)	59.70 ^b^ (0.40)	1.829 ^d^ (0.042)
Whole 25%	17.53 ^c^ (0.28)	79.69 ^b^ (0.29)	61.02 ^b^ (0.21)	2.366 ^c^ (0.064)
Whole 50%	21.61 ^b^ (0.68)	80.96 ^b^ (0.54)	61.58 ^b^ (0.74)	2.675 ^bc^ (0.102)
Whole 100%	28.27 ^a^ (0.28)	82.49 ^a^ (0.26)	64.15 ^a^ (0.40)	2.928 ^b^ (0.058)
Fine 25%	18.16 ^c^ (0.30)	82.70 ^a^ (0.35)	64.95 ^a^ (0.67)	2.937 ^b^ (0.089)
Fine 50%	21.87 ^b^ (0.23)	82.54 ^a^ (0.30)	64.05 ^a^ (0.34)	2.957 ^b^ (0.073)
Fine 100%	29.52 ^a^ (0.20)	83.89 ^a^ (0.22)	64.92 ^a^ (0.32)	3.364 ^a^ (0.077)

N = 63. Means (SE) not followed by the same letter within the same column are significantly different (*p* < 0.05).

**Table 3 foods-12-03004-t003:** ΔE* results of the CIELAB values between every tofu sample with added okara.

	ΔE*						
	Control	Whole 25%	Whole 50%	Whole 100%	Fine 25%	Fine 50%	Fine 100%
Control	0	1.129	1.711	2.338	0.950	2.394	1.071
Whole 25%		0	0.630	1.379	1.467	1.905	0.557
Whole 50%			0	0.805	1.840	2.171	0.872
Whole 100%				0	2.358	2.790	1.551
Fine 25%					0	3.091	1.031
Fine 50%						0	2.326

**Table 4 foods-12-03004-t004:** Result of the texture profile analysis for tofu with added okara.

Sample	Hardness (g)	Resilience	Chewiness	Springiness	Cohesiveness
Control	855 ^abc^ (18.7)	0.175 ^ab^ (0.009)	98.9 ^abc^ (19.0)	0.382 ^a^ (0.062)	0.289 ^abc^ (0.017)
Whole 25%	1047 ^ab^ (28.9)	0.177 ^ab^ (0.007)	118.4 ^ab^ (7.52)	0.373 ^a^ (0.023)	0.304 ^abc^ (0.005)
Whole 50%	1315 ^a^ (32.3)	0.117 ^c^ (0.004)	133.8 ^a^ (9.92)	0.324 ^a^ (0.010)	0.311 ^abc^ (0.011)
Whole 100%	875 ^ab^ (49.7)	0.162 ^b^ (0.004)	108.7 ^ab^ (14.3)	0.350 ^a^ (0.016)	0.341 ^a^ (0.012)
Fine 25%	523 ^c^ (23.1)	0.202 ^a^ (0.015)	42.17 ^c^ (4.11)	0.284 ^a^ (0.014)	0.280 ^bc^ (0.010)
Fine 50%	796 ^bc^ (14.1)	0.101 ^c^ (0.002)	65.36 ^bc^ (3.57)	0.280 ^a^ (0.032)	0.270 ^c^ (0.009)
Fine 100%	924 ^ab^ (121.7)	0.106 ^c^ (0.006)	107.9 ^ab^ (24.3)	0.324 ^a^ (0.014)	0.326 ^ab^ (0.018)

N = 63. Means (SE) not followed by the same letter within the same column are significantly different (*p* < 0.05).

**Table 5 foods-12-03004-t005:** Storage modulus (G′) and tan δ during the gelation of soy milk with added okara (see Figure 4 for the reference points).

	T_0_		T_1_		T_2_		T_3_	
	G′	Tan δ	G′	Tan δ	G′	Tan δ	G′	Tan δ
Control	5.73 ^d^ (0.39)	0.565 ^a^ (0.014)	8.26 ^e^ (0.46)	0.324 ^a^ (0.019)	20.69 ^d^ (1.90)	0.158 ^a^ (0.018)	40.63 ^d^ (4.05)	0.177 ^a^ (0.009)
Whole 25%	8.89 ^cd^ (0.59)	0.503 ^b^ (0.009)	14.50 ^cd^ (0.64)	0.295 ^ab^ (0.012)	32.80 ^bcd^ (1.44)	0.136 ^ab^ (0.006)	67.36 ^bc^ (2.12)	0.170 ^ab^ (0.003)
Whole 50%	13.06 ^ab^ (1.28)	0.446 ^cd^ (0.016)	20.98 ^ab^ (2.19)	0.242 ^bc^ (0.013)	45.32 ^b^ (3.82)	0.113 ^bc^ (0.007)	91.92 ^ab^ (7.56)	0.162 ^ab^ (0.003)
Whole 100%	14.29 ^a^ (1.24)	0.418 ^d^ (0.013)	19.82 ^abc^ (1.91)	0.231 ^cd^ (0.012)	39.70 ^bc^ (3.81)	0.105 ^bc^ (0.006)	73.31 ^bc^ (7.26)	0.161 ^ab^ (0.003)
Fine 25%	7.39 ^cd^ (0.71)	0.482 ^bc^ (0.012)	11.83 ^de^ (1.02)	0.260 ^bc^ (0.014)	27.65 ^cd^ (1.65)	0.121 ^abc^ (0.008)	58.02 ^cd^ (3.09)	0.166 ^ab^ (0.004)
Fine 50%	9.99 ^bc^ (0.64)	0.412 ^d^ (0.010)	16.51 ^bcd^ (1.11)	0.213 ^cd^ (0.007)	39.18 ^bc^ (2.73)	0.094 ^c^ (0.004)	77.53 ^bc^ (5.07)	0.156 ^b^ (0.002)
Fine 100%	17.18 ^a^ (1.34)	0.354 ^e^ (0.004)	24.25 ^a^ (1.58)	0.182 ^d^ (0.005)	61.87 ^a^ (7.04)	0.089 ^c^ (0.004)	108.39 ^a^ (9.13)	0.157 ^b^ (0.001)

T_0_: initial soy milk at 25 °C; T_1_: sample after heating to 85 °C (5 °C/min); T_2_: sample after isothermal hold at 85 °C for 60 min.; T_3_: sample after final cooling to 25 °C (5 °C/min); N = 63. Data expressed as the mean (standard error). Values not followed by the same letter within the same column are significantly different (*p* < 0.05).

**Table 6 foods-12-03004-t006:** Estimated nutrient compositions of okara-fortified tofu (calculations based on values determined by Van Der Riet et al. 1989 [11]).

Okara % ^a^	Protein (g/100 g)	Oil (g/100 g)	Carbohydrate (g/100 g)	Total Fiber (g/100 g)	Ash (g/100 g)
Control	53.9	30.2	3.40	5.40	7.20
25%	49.6	26.7	3.72	13.9	6.53
50%	46.6	24.3	3.95	20.2	6.05
100%	41.8	20.4	4.30	29.6	5.31

^a^ Sufficient okara added to give 0, 25, 50, or 100% of the amount recovered from the original soybeans (that is, 0, 7.7, 15.4, or 30.7 g okara per 100 g hulled soybeans).

## Data Availability

Data is contained within the article.

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
