# Peer review of "The Effects of Okara Ratio and Particle Size on the Physical Properties and Consumer Acceptance of Tofu"

_foods, 2023, doi:10.3390/foods12163004_

Round 1

Reviewer 1 Report

This article discusses The Effects of Okara Ratio and Particle Size on the Physical Properties and Consumer Acceptance of Tofu. However, some things need to be corrected in this manuscript:

1.       It is necessary to add to the abstract section related to research methods and results related to particle size and consumer acceptance of tofu.

2.       Title without a period.

3.       Figure 1 has 2 in line 53 and line 322.

4.       Preliminary research writing should be rewritten to clarify the flow of problems to be studied. The introductory flow needs to be written systematically and show the urgency and novelty of the research. In the introduction, there is also no need to use process diagrams.

5.       Lines 90-92 need to be equipped with specific quantitative data on the primary raw materials used in the research.

6.       Line 94 (Tofu and Okara Production), providing a process flow chart to clarify the description is necessary.

7.       Line 134, writing the table title is placed above the Table. (check for all tables)

8.       Line 221 needs to be equipped with a texture profile curve image showing the texture parameters being analyzed.

9.       In general, the discussion could be more in-depth and comprehensive, primarily related to the phenomenon of the resulting data.

10.   The writing format should be corrected according to the template guide provided.

Extensive editing of English language required

Author Response

Modifications to address the concerns of Reviewer #1 are presented in RED

This article discusses The Effects of Okara Ratio and Particle Size on the Physical Properties and Consumer Acceptance of Tofu. However, some things need to be corrected in this manuscript:

  1. It is necessary to add to the abstract section related to research methods and results related to particle size and consumer acceptance of tofu.
    The journal’s style guide limits the length of abstracts to 200 words. The originally submitted version had a 190-word abstract. While we have attempted to include some information related to the subjects you suggest, we were unable to fully describe everything due to the limitations.

  1. Title without a period.
    We assume that you are suggesting that we remove the period from the title, and have done so

  1. Figure 1 has 2 in line 53 and line 322.

  1. Preliminary research writing should be rewritten to clarify the flow of problems to be studied. The introductory flow needs to be written systematically and show the urgency and novelty of the research. In the introduction, there is also no need to use process diagrams.
    We have relocated the process diagram to the Materials and Methods section, as you requested below.
  2. Lines 90-92 need to be equipped with specific quantitative data on the primary raw materials used in the research.
    The authors are uncertain what the reviewer means by this concern. The section referenced details the type of materials, and the specific amounts used are referenced in the subsequent section on Soymilk and Okara Production.

  1. Line 94 (Tofu and Okara Production), providing a process flow chart to clarify the description is necessary.

  1. Line 134, writing the table title is placed above the Table. (check for all tables)

Corrected

  1. Line 221 needs to be equipped with a texture profile curve image showing the texture parameters being analyzed.

We have added an image that we believe will satisfy the reviewer’s concern.

  1. In general, the discussion could be more in-depth and comprehensive, primarily related to the phenomenon of the resulting data.

 The discussion section has been greatly expanded and revised to address reviewer comments.

  1. The writing format should be corrected according to the template guide provided.

We believe the manuscript is now properly formatted, but without any specific examples, we cannot determine whether your concerns have been addressed.

Reviewer 2 Report

In this paper, the effects of ratio and particle size of Okara on the physical properties of tofu were investigated. In whole, this processing method increased the utilization and economic value of by-products, and to some degree, the finial product appeal to the consumer by adjusting ratio and particle of Okara. The work as a whole is well articulated and seems interesting. Some minor points should be improved as described below.

- Fig.1, this picture is not clear, please enhance the resolution.
- For the color measurement, please provide the parameters (L, a, b) of white plate and the determination temperature.

- Table 5, please keep the consistence of group name, for instance, change W25% with Whole 25%.

Author Response

Modifications to address the concerns of Reviewer #2 are presented in BLUE

In this paper, the effects of ratio and particle size of Okara on the physical properties of tofu were investigated. In whole, this processing method increased the utilization and economic value of by-products, and to some degree, the finial product appeal to the consumer by adjusting ratio and particle of Okara. The work as a whole is well articulated and seems interesting. Some minor points should be improved as described below.

- Fig.1, this picture is not clear, please enhance the resolution.

We have created an improved image, at a higher resolution, and replaced the old, unclear image.

- For the color measurement, please provide the parameters (L, a, b) of white plate and the determin
ation temperature.

We have included the reference values for the calibration plate, and indicated that calibration was performed at room temperature. We do not have a recorded value for that temperature, but it as the building is climate controlled, it would have been between 19 and 24 °C

- Table 5, please keep the consistence of group name, for instance, change W25% with Whole 25%.

We have corrected all tables with inconsistent names.

Reviewer 3 Report

I am very grateful you for the invitation to review manuscript foods-2481136 by Joo and coauthors "The Effects of Okara Ratio and Particle Size on the Physical Properties and Consumer Acceptance of Tofu”. This study examined the effects of adding different levels of the soymilk byproduct okara, with either fine (<180 μm) or whole particles, on the physical properties of tofu coagulated with nigari. The work is interesting but needs adjustments to increase the quality of the material.

Comments:

- Abstract: Include a brief sentence about the importance of the study (problem to be solved).

- Line 15: Specify the term nigari.

- Abstract: Please indicate in the abstract a brief and better step-by-step about the work including the parameters and conditions used.

- Abstract: Present the most specific results. Insert numerical results related to the main findings of the work.

- Line 27-28: Change the repeated keywords by different words from the title.

- Line 32: Change “year 965 [1] Due” to “year 965 [1]. Due”.

- Introduction: Include production, consumption, and market information for tofu.

- Line 96: Properly specify the modifications.

- Line 134-135: Insert table title above the table.

- Table 1: The determination or estimation of the components must be better approached, mainly considering that each okara or soybean can present variable compositions.

- Results: The calculated proximate compositions should be better explained, or removed, since there is a large variation in relation to the soybean data.

- Line 284: Calculations based on data from van der Riet et al. [11] should be better explained. Was the data calculated based on the technical data of the product presented by the manufacturers?

- Lines 285-294: Authors should better explain what was done at this point.

- Line 294 and others: Table title must appear before the table.

- Figure 3: improve resolution.

- Line 484: Please explain in more detail the role of the fiber and bonding and absorption with water.

- Discussion: Should be improved regarding the chemical interactions of components in tofu.

- Lines 491-494: Higher fiber concentration impacts protein binding and, consequently, water release. Better detail the content.

- Line 509, 4.2 Color Analysis: Authors must correlate the color parameter to its components and interactions. The item is generic.

- Line 519, 4.3 Texture Profile Analysis: The item is superficial and needs theoretical deepening and a detailed discussion.

Author Response

Modifications to address the concerns of Reviewer #3 are presented in GREEN

- Abstract: Include a brief sentence about the importance of the study (problem to be solved).

We have revised the abstract to better highlight the overall problem, but, as with our response to reviewer #1, we were constrained by the journal’s word-count limitations.

- Line 15: Specify the term nigari.

We revised the Abstract to indicate nigari is a traditional coagulant.

- Abstract: Please indicate in the abstract a brief and better step-by-step about the work including the parameters and conditions used.

As best we could, given the 200-word limitation, we have attempted to do so.

- Abstract: Present the most specific results. Insert numerical results related to the main findings of the work.

As best we could, given the 200-word limitation, we have attempted to do so.

- Line 27-28: Change the repeated keywords by different words from the title.

We thank the reviewer for their concern and did consider removing the keywords “Tofu” and “Okara”, as that would still give us four keywords (only 3 are required). However, after looking at the most recently published articles in the journal, it appears that there is a significant overlap in keywords and words used in the title. Thus, unless you feel particularly strongly about this, we would prefer to keep the 2 keywords.

- Line 32: Change “year 965 [1] Due” to “year 965 [1]. Due”.

Correction made.

- Introduction: Include production, consumption, and market information for tofu.

- Line 96: Properly specify the modifications.

In the original text, this was explained in the subsequent lines.

- Line 134-135: Insert table title above the table.

Corrected

- Table 1: The determination or estimation of the components must be better approached, mainly considering that each okara or soybean can present variable compositions.

We have re-titled the table to address potential misunderstandings, and have moved it to the discussion section (as Table 6). While we agree that there can be variability due to cultivar, growing conditions, etc, actual analysis of the specific materials used was far outside both the scope and budget of this work. We did, however, feel that the information was something a reader might be interested in, and could help explain some of the observed phenomena    

- Results: The calculated proximate compositions should be better explained, or removed, since there is a large variation in relation to the soybean data.

We have relocated this section to the discussion section.

- Line 284: Calculations based on data from van der Riet et al. [11] should be better explained. Was the data calculated based on the technical data of the product presented by the manufacturers?

The tofu in this study was manufactured by the authors, and thus did not come with technical data. As mentioned above, we recognize that in an ideal world, proximate analysis would have been performed on our samples, but again, this was outside of the scope of this work.

- Lines 285-294: Authors should better explain what was done at this point.

- Line 294 and others: Table title must appear before the table.

Corrected

- Figure 3: improve resolution.

We have provided a better version of the image.

- Line 484: Please explain in more detail the role of the fiber and bonding and absorption with water.
- Discussion: Should be improved regarding the chemical interactions of components in tofu.
- Lines 491-494: Higher fiber concentration impacts protein binding and, consequently, water release. Better detail the content.

This section was significantly revised, with additional references, etc.

- Line 509, 4.2 Color Analysis: Authors must correlate the color parameter to its components and interactions. The item is generic.

This section was significantly revised, with additional references, etc.

- Line 519, 4.3 Texture Profile Analysis: The item is superficial and needs theoretical deepening and a detailed discussion.

This discussion session underwent extensive revision to hopefully address the reviewer’s concerns. New references and a greater depth of discussion were added, in addition to general revisions for improved clarity.

Reviewer 4 Report

The authors studied the effect of okara addition on the properties of tofu. The study is well-designed and provided some interesting results to readers. However, some points has to be addressed to improve the manuscript.

1. The table head should be placed before the table. 

2. 2.8 sensory evaluation. the purchase desire evaluation is not objective. At the actually supermarket, there will not be someone to present the nutritional and environmental advantages of okara. So the result of purchase ratio in Figure 4 is not reliable. 

3.  Table 5, there is a comment on the table. Please check the format for the whole manuscript. 

4. The picture of the tofu should be added to section 3.2 color analysis. 

5. References should be added with some recent manuscripts, especially in the introduction part to support the novelty of the study. 

Author Response

Modifications to address the concerns of Reviewer #1 are presented in VIOLET

The authors studied the effect of okara addition on the properties of tofu. The study is well-designed and provided some interesting results to readers. However, some points has to be addressed to improve the manuscript.

  1. The table head should be placed before the table. 

We have corrected this throughout the manuscript.

  1. 2.8 sensory evaluation. the purchase desire evaluation is not objective. At the actually supermarket, there will not be someone to present the nutritional and environmental advantages of okara. So the result of purchase ratio in Figure 4 is not reliable. 
    While we understand this concern, we believe that the required Nutritional Information Panel, potentially coupled with on-package advertising would present the potential consumer with the relevant information. We have amended the relevant discussion section and included additional references, including some more recent ones.
  2. Table 5, there is a comment on the table. Please check the format for the whole manuscript. 

Apologies for this- we had addressed the comment and thought we had deleted it, but something must have gone awry. I believe it may have been an Apple vs PC issue, but cannot be certain.

  1. The picture of the tofu should be added to section 3.2 color analysis. 

The authors are concerned that, due to the limitations of the equipment used to photograph the samples along with the size constraints of the publication, any perceptible differences between the samples would be lost. If this is something the reviewer feels strongly about, we can include them, but at the moment, the co-author who has them is out of country.

  1. References should be added with some recent manuscripts, especially in the introduction part to support the novelty of the study. 

We have added some more recent references

Round 2

Reviewer 1 Report

The manuscript has been revised and is better than before, so it is worthy of publication.